# Treatment of Anti-HLA Donor-Specific Antibodies Results in Increased Infectious Complications and Impairs Survival after Liver Transplantation

**DOI:** 10.3390/jcm9123986

**Published:** 2020-12-09

**Authors:** Sinem Ünlü, Nils Lachmann, Maximilian Jara, Paul Viktor Ritschl, Leke Wiering, Dennis Eurich, Christian Denecke, Matthias Biebl, Sascha Chopra, Safak Gül-Klein, Wenzel Schöning, Moritz Schmelzle, Petra Reinke, Frank Tacke, Johann Pratschke, Robert Öllinger, Tomasz Dziodzio

**Affiliations:** 1Department of Surgery, Campus Charité Mitte and Campus Virchow-Klinikum, Charité-Universitätsmedizin Berlin, 13353 Berlin, Germany; sinem.uenlue@charite.de (S.Ü.); maximilian.jara@charite.de (M.J.); paul.ritschl@charite.de (P.V.R.); leke.wiering@charite.de (L.W.); dennis.eurich@charite.de (D.E.); christian.denecke@charite.de (C.D.); matthias.biebl@charite.de (M.B.); sascha.chopra@charite.de (S.C.); safak.klein@charite.de (S.G.-K.); wenzel.schoening@charite.de (W.S.); moritz.schmelzle@charite.de (M.S.); johann.pratschke@charite.de (J.P.); robert.oellinger@charite.de (R.Ö.); 2Institute for Transfusion Medicine, H&I Laboratory, Charité-Universitätsmedizin Berlin, 13353 Berlin, Germany; nils.lachmann@charite.de; 3BIH Charité Clinician Scientist Program, Berlin Institute of Health (BIH), 10178 Berlin, Germany; 4Department of Nephrology and Internal Intensive Medicine, Charité-Universitätsmedizin Berlin, 13353 Berlin, Germany; petra.reinke@charite.de; 5Department of Hepatology and Gastroenterology, Charité-Universitätsmedizin Berlin, 13353 Berlin, Germany; frank.tacke@charite.de

**Keywords:** DSA, survival analysis, sepsis, rejection treatment

## Abstract

Donor-specific anti-human leukocyte antigen antibodies (DSA) are controversially discussed in the context of liver transplantation (LT). We investigated the relationship between the presence of DSA and the outcome after LT. All the LTs performed at our center between 1 January 2008 and 31 December 2015 were examined. Recipients < 18 years, living donor-, combined, high-urgency-, and re-transplantations were excluded. Out of 510 LTs, 113 DSA-positive cases were propensity score-matched with DSA-negative cases based on the components of the Balance of Risk score. One-, three-, and five-year survival after LT were 74.3% in DSA-positive vs. 84.8% (*p* = 0.053) in DSA-negative recipients, 71.8% vs. 71.5% (*p* = 0.821), and 69.3% vs. 64.9% (*p* = 0.818), respectively. Rejection therapy was more often applied to DSA-positive recipients (*n* = 77 (68.1%) vs. 37 (32.7%) in the control group, *p* < 0.001). At one year after LT, 9.7% of DSA-positive patients died due to sepsis compared to 1.8% in the DSA-negative group (*p* = 0.046). The remaining causes of death were comparable in both groups (cardiovascular 6.2% vs. 8.0%; *p* = 0.692; hepatic 3.5% vs. 2.7%, *p* = 0.788; malignancy 3.5% vs. 2.7%, *p* = 0.788). DSA seem to have an indirect effect on the outcome of adult LTs, impacting decision-making in post-transplant immunosuppression and rejection therapies and ultimately increasing mortality due to infectious complications.

## 1. Introduction

The clinical relevance of donor-specific anti-human leukocyte antigen antibodies (DSA) in liver transplantation (LT) has been the basis for many controversial discussions. In kidney transplantation, the negative effects of preformed and de novo DSA on patient and graft survival have been well demonstrated [1,2]. Furthermore, the presence of DSA in other solid organ transplantations, such as of the lung [3], heart [4], or pancreas [5], has been reported to be associated with inferior graft outcomes. For many years, liver grafts have been considered less vulnerable to DSA due to the graft size, dual blood supply, and the patient’s own immunological activity [6]. Since the first observations of antibody-mediated rejections (ABMR) in LT 30 years ago [7], especially recent data have led to a new perception of DSA in the context of LT. In 2016, the Banff Working Group provided a first approach on standardized (histopathological) ABMR criteria [8] and new laboratory techniques, such as the Luminex^®^ assay, helping to achieve a more precise antibody detection, specification, and quantification [9,10]. Additionally, pathologic conditions such as T-cell-mediated rejections (TCMR) and infections can lead to an upregulation of tissue human leukocyte antigen (HLA) expression and make the liver graft more susceptible to ABMR [6,11,12].

Recent findings indicate an association between DSA and early/chronic rejections and graft injury [13,14,15,16,17,18] However, the data regarding the impact of DSA on patient and graft survival after LT are less clear [13,14,15,16,18,19,20,21,22,23].

Consequently, there is still a need for data to clarify the effects of DSA’s presence on LT outcomes. The aim of this study was to investigate the impact of DSA on patient and graft survival by means of a matched case-control analysis and to identify risk factors for inferior patient and graft outcomes.

## 2. Patients and Methods

### 2.1. Patient Recruitment and Study Design

Since January 2008, DSA were prospectively assessed by the Luminex^®^ assay in all patients waitlisted for LT and post-LT. All patients undergoing deceased organ donor LT at the Department of Surgery, Campus Charité Mitte and Campus Virchow-Klinikum, Berlin, Germany, from 1 January 2008 to 31 December 2015, were examined, with follow-up ceasing on 1 January 2018. Combined liver-kidney, multi-visceral, high-urgency [24], and re-transplantations or patients who were under the age of 18 years at the time of LT were excluded from the analysis (Figure 1). 

The study cohort was divided into two groups (DSA-positive and DSA-negative patients) and compared regarding their demographic variables and transplant outcomes. A 1:1 propensity score matching of DSA-positive and DSA-negative individuals based on the components of the BAR (Balance of Risk) score was performed and the groups were compared. The primary endpoints were patient and graft survival regarding the presence of DSA. Secondary endpoints were the appearance of any rejection and cause of death. The study was approved by the institutional ethics committee (ID: EA4/061/17).

### 2.2. Data Collection and Definition of Patient and Graft Survival

Electronic records of recipient information were obtained from a prospectively collected hospital database (SAP^®^ SE, Walldorf, Germany). Anonymous donor data were acquired from the Eurotransplant Network Information System (ENIS). 

Cold ischemia time (CIT), warm ischemia time (WIT), patient survival, and graft survival were defined according to the United Network for Organ Sharing (UNOS) criteria [25,26]. 

### 2.3. Antibody Screening

The detection and specification of anti-HLA antibodies were performed using LABScreen^®^ Mixed and Single Antigen Beads (OneLambda, West Hills, CA, USA), respectively, according to the authors’ previously published work [9]. Samples were measured on a Luminex^®^ 200 (Luminex^®^, Austin, TX, USA) and analyzed using the HLA Fusion software (OneLambda). Donor-HLA-typing for HLA-A, -B, -C, -DRB1, and -DQB1 was acquired from ENIS and matched with the antibody specificities of the recipient to define the DSA. Organ recipients were routinely screened for DSA before LT listing and on the day of LT to detect preformed DSA. Postoperatively, de novo DSA screening was performed on a weekly basis until the discharge from the ICU and at the request of the treating surgeon. During routine check-ups, DSA detection was performed in patients with graft function impairment or for the follow-up of previously detected DSA. Preformed DSA were defined as antibodies present before or at the time of LT. De novo DSA were defined as newly detected antibodies after LT or antibodies against epitopes that were not present before LT in patients with preformed DSA. Data were normalized to negative control serum and all the DSA exceeding a mean fluorescence intensity (MFI) of 1000 were considered positive. 

### 2.4. Postoperative Management 

Liver function parameters were monitored daily during the entire perioperative stay and routinely examined after 3, 6, 12, 18, 24, 36, and 60 months. Postoperative care and immunosuppression (IS) regimens were provided according to a standardized protocol. In the case of LTs due to autoimmune liver disease (AILD), antibody induction with antithymocyte globulin (ATG) or IL-2 receptor antagonists was applied. The standard IS consisted of prednisolone and tacrolimus. No adaptation of IS has been made in the case of preformed DSA. In AILD recipients, the IS was extended by mycophenolic acid. Initial tacrolimus levels were set at 6–8 ng/mL in standard LTs and 8–10 ng/mL in LTs due to AILD and tapered within the first 6 months to maintenance levels of 4–6 ng/mL in all patients. Steroids were tapered off for most patients within 2 months after LT. Patients who underwent LT due to AILD were maintained on a double therapy with tacrolimus and mycophenolate. 

### 2.5. Rejection Diagnosis and Treatment

Biopsies were performed in case of suspected rejection and routinely at 1, 3, and 5 years post LT. Histological diagnosis and the grading of allograft rejections was classified according to the Banff criteria of 2000 and later its update of 2016 [8,27]. Rejection therapy was initiated in the case of histologically proven rejection or in patients with a suspected rejection due to a significant increase in liver function parameters and after the exclusion of other causes. Initial rejection therapy consisted of steroids and increased doses of the standard IS. In ABMR, the initial immunosuppressive therapy was augmented by plasmapheresis (PP), intravenous immunoglobulins (IVIGs), and occasionally rituximab. Refractory TCMR were treated with ATG. Cases receiving rejection treatment with equivocal histological findings or without a bioptical verification of rejection were retrospectively defined as ABMR or TCMR treated “on clinical suspicion” depending on the therapy received. 

### 2.6. Survival Stratification Model and Outcome

The BAR score achieves the most accurate outcome stratification in terms of patient survival compared to all known scores and was used for the outcome stratification and matching process [28,29,30]. The BAR score and MELD were calculated according to the UNOS and Eurotransplant formulas [28,30] and the donor risk index (DRI) according to the published formula by Feng et al. [31]. 

### 2.7. Statistical Analyses

Statistical analyses were carried out using IBM SPSS Statistics, version 25 (IBM Corporation, Armonk, NY, USA), and GraphPad Prism, version 6.01 (GraphPad Software, Inc., San Diego, CA, USA). Categorical data are presented as frequencies and percentages and were compared by Pearson’s chi-squared test. Continuous data are presented as median and interquartile range and compared by the non-parametric Mann–Whitney U test. The propensity score method with a logistic model was used to match cases and controls. The matching process included the following components of the BAR score: MELD score, donor age, recipient age, pretransplant mechanical, ventilated or organ-perfusion support, and CIT. Retransplantation was an exclusion criterion for the analysis and was not embedded in the matching process. 

Patient and organ survival were analyzed by the Kaplan–Meier method and the log-rank test to compare groups. A two-sided *p*-value of <0.05 was considered statistically significant. 

## 3. Results

### 3.1. Study Population

During the observation period, a total of 687 LTs were performed. The final analysis comprised 510 patients. Clinical data were available for all patients. DSA were identified in 113 LT recipients (22.2%). The remaining 397 patients (77.8%) were DSA-negative. Within the DSA-positive LT recipients, 49 (43.4%) individuals showed preformed, 55 (48.6%) de novo, and 9 (8%) individuals were tested positive for both preformed and de novo DSA. The median time to the first detection of de novo DSA was 13 days (9.5–20). Class II DSA were present in 50.9% (*n* = 28) of the patients with de novo and 10.2% of those with preformed DSA (*n* = 5, *p* < 0.001) (data listed in Appendix A).

The median follow-up was 58 (29–86) months. In the unmatched analysis, the median recipient age at LT was 54 (48–61) years in the DSA-positive group and 57 (51–62) years in the DSA-negative group (*p* = 0.073). The main indication of LT in both groups DSA-positive vs. DSA-negative was alcoholic cirrhosis (*n* = 34 (30.1%) vs. *n* = 188 (47.4%); *p* = 0.001), followed by hepatocellular carcinoma (*n* = 32 (28.3%), vs. *n* = 157 (39.5%), *p* = 0.029). The median BAR score at LT significantly differed between the DSA-positive and the DSA-negative group (8 (3–13) vs. 6 (3–9), *p* = 0.013). There was no significant difference in the MELD score between both groups (16 (9–22) in DSA-positive vs. 15 (10–20) in DSA-negative recipients, *p* = 0.377; Table 1). Out of all DSA-negative patients, 28% (*n* = 111) showed a female-to-male (F-M) donor-to-recipient mismatch. The DSA-positive group consisted of 17.7% (*n* = 20, *p* = 0.028) F-M mismatches. 

#### BAR Score Group Matching

Out of the 397 DSA-negative individuals, 113 patients were matched 1:1 with 113 DSA-positive individuals (Table 2). Median follow-up was 59 (25–101) months.

In both groups, the main indication for LT remained alcoholic cirrhosis (DSA-positive: *n* = 34 (30.1%) vs. 56 (49.6%) DSA-negative, *p* = 0.003) and hepatocellular carcinoma (DSA-positive *n* = 32 (28.3%) vs. 45 (39.8%) DSA-negative, *p* = 0.068). Organ recipients with PSC/PBC/AIH and NASH as underlying disease were more frequently represented in the DSA-positive group. No significant differences were observed between the DSA-positive and DSA-negative group regarding the BAR score (8 (3–13) vs. 7 (4–13), *p* = 0.619) and the distribution of its single components (MELD at LT: 16 (9–22) vs. 16 (10–29), *p* = 0.426; necessity of pretransplant mechanical, ventilated or organ-perfusion support: *n* = 6 (5.4%) vs. 4 (3.6%), *p* = 0.322; recipient age: 54 (48–61) vs. 57 (51–62) years, *p* = 0.061; and donor age: 56 (44–69) vs. 59 (45–70) years, *p* = 0.301). The number of high labMELD (> 35) patients was comparable in both groups (*n* = 20 (17.7%) vs. 16 (14.2%), *p* = 0.686).

DSA-positive recipients showed a shorter CIT and WIT compared to the DSA-negative recipients (CIT: 9.7 (8–11) vs. 10.2 (9–12) hours, *p* = 0.021, and WIT: 43 (38–50) vs. 48 (40–56) minutes; *p* = 0.012).

No differences in the overall courses of liver function parameters were observed between the groups, apart from only three particular lab tests being statistically significant at distinct points in time (Gamma-glutamyl transferase: on POD 7: DSA-group 245 (134–435) vs. 206 (118–308) in the control group, *p* = 0.047; Albumin on POD 7: 2.9 (2.7–3.1) vs. 3.0 (2.8–3.3), *p* = 0.010; and alkaline phosphatase 12 months after LT: 106 (81–202) vs. 97 (75–129), *p* = 0.043; Figure 2). 

The median length of stay (38 (28–62) vs. 32 (23–49) days; *p* = 0.012) and median time spent on ICU after LT (13 (7–32) vs. 9 (6–15) days; *p* = 0.008) were both significantly longer in the DSA-positive patients. No significant difference was observed with regard to the F-M donor-to-recipient mismatch between the groups (DSA-positive group *n* = 20 (17.7%) vs. *n* = 30 (26.6%) in the DSA-negative group, *p* = 0.109).

### 3.2. Patient and Graft Survival

#### 3.2.1. DSA-Positive vs. All DSA-Negative LT Recipients

The overall patient and graft survival is shown in Figure 3a. In the unmatched analysis at one year, the patient survival after LT was significantly lower in DSA-positive compared to DSA-negative allograft recipients (74.4% vs. 86.3%; *p* = 0.003). No significant differences in patient survival were observed at three (71.8% vs. 76.9%; *p* = 0.156) and five years after LT (69.3% vs. 71.8%; *p* = 0.400). The graft survival of DSA-positive individuals was inferior at one year after LT compared to DSA-negative individuals (73.4% vs. 80.2%; *p* = 0.167), with no significant difference at three (68.9% vs. 70.8%; *p* = 0.636) and five years (66.6% vs. 64.7%; *p* = 0.636) after LT.

#### 3.2.2. DSA-Positive vs. Matched DSA-Negative LT Recipients

In the matched analysis, the one-year patient survival after LT was 74.4% in the DSA-positive vs. 84.8% (*p* = 0.053) in the DSA-negative cohort, 71.8% vs. 71.2% (*p* = 0.821) at three years and 69.3% vs. 64.9% (*p* = 0.818) at five years, respectively. Infections and sepsis were the major causes of death in the DSA-positive allograft recipients and significantly more frequent than in the matched DSA-negative cohort (*n* = 11 (9.7%) vs. *n* = 2 (1.8%); *p* = 0.046). Hospital-acquired pneumonia was the main cause of sepsis (*n* = 7 in DSA-positive; and *n* = 2 in DSA-negative patients). All the infection-related deaths occurred within the first 6 months after LT. The distribution of the remaining causes of death was comparable in both groups (cardiovascular 6.2% vs. 8.0%; *p* = 0.692; hepatic 3.5% vs. 2.7%, *p* = 0.788; malignancy 3.5% vs. 2.7%, *p* = 0.788). DSA-positive individuals showed an inferior one-year graft survival compared to the matched DSA-negative ones (73.4% vs. 79.5%; *p* = 0.270), with a non-significant reciprocal trend at five years (66.5% vs. 57.7%; *p* = 0.339). Retransplantations were necessary in 6 (5.3%) DSA-positive and 11 (9.7%) DSA-negative LT recipients (*p* = 0.207).

Significantly more patients were female (48.7%) in the DSA-positive group than in the matched DSA-negative control (29.2%, *p* = 0.003). The proportion was higher in LT recipients with preformed DSA (65.3% women).

#### 3.2.3. DSA Subgroup Analysis

Recipients with preformed DSA showed an inferior one-year patient (67.4% vs. 84.8%; *p* = 0.008) and graft survival (65.3% vs. 79.5%; *p* = 0.062) compared to the matched DSA-negative individuals, with sepsis as the leading cause of death (preformed DSA: *n* = 7 (6.2%) vs. *n* = 2 (1.8%); *p* = 0.031; Table 3). The lower survival rate in the first year was even more pronounced in patients with preformed DSA and a MFImax of ≥5000 (*n*= 19, 63.2% vs. 84.8%, *p* = 0.022). No significant differences in mortality or graft loss were observed between individuals with de novo and matched DSA-negative individuals at any time point (Figure 3b), regardless of the MFI level.

### 3.3. Rejections and Rejection Treatment

Within the first year after LT, biopsies were performed in 72 DSA-positive (63.7%) and 127 DSA-negative patients (32.0%, *p* < 0.001). Rejection therapies were more frequently conducted in the DSA-positive (*n* = 77, 68.1%) than in the DSA-negative group (*n* = 135, 34.0%, *p* < 0.001; Table 4). In the matched cohorts, ABMR were histologically confirmed in 2 (1.8%) DSA-positive cases and no DSA-negative cases (*p* = 0.155). ABMR treatment was applied in 41 (36.3%) DSA-positive and in 2 (1.8%, *p* < 0.001) DSA-negative LT recipients. TCMR were histologically confirmed in 59 (52.2%) DSA-positive and 34 (30.1%, *p* < 0.001) DSA-negative recipients. In total, 71 (62.8%) DSA-positive and 37 (32.7%, *p* < 0.001) DSA-negative individuals received TCMR therapy. The most common rejection therapy was a combination of steroids and increase in the standard IS dose (DSA-positive group: *n* = 70 (61.9%) vs. 37 (32.7%) DSA-negative group, *p* < 0.001). The overall number of treated ABMR and TCMR was comparable in the LT recipients with preformed and de novo DSA (Table 5). Patients with de novo DSA were more often treated for biopsy-proven TCMR (*n* = 35, 63.6%) than patients with preformed DSA (*n* = 19, 38.8%, *p* = 0.011) and without DSA (*n* = 34, 30.1%, *p* < 0.001). Six DSA-positive (5.3%) and 3 DSA-negative patients (2.7%, *p* = 0.307) were treated for TCMR on clinical suspicion without histological verification. In two (1.8%) of these DSA-positive patients, rejection therapy was additionally augmented by PP/IVIG due to suspected refractory rejection and the presence of preformed DSA. In one DSA-positive patient (0.9%) with preformed DSA, PP/IVIG therapy was applied without biopsy. 

Patients with preformed DSA received the first rejection therapy significantly earlier compared to patients with de novo DSA (6 (5–16) vs. 11.5 (7–18) days; *p* = 0.024) and DSA-negative patients (9 (7–23) days, *p* = 0.003).

## 4. Discussion

This study demonstrates that especially preformed DSA are associated with an impaired (one-year) patient survival after LT, and sepsis-related mortality needs to be considered as a major cause. No remarkable adverse effects on graft survival were observed. Our data thereby indicate that DSA presence may have an indirect effect on the outcome in adult LTs by interfering with the diagnosis of rejections and the decision-making in post-transplant IS and rejection therapies.

The incidence of DSA (22.2%) in our study is within the range of the published data. We observed 9.6% preformed and 10.8% de novo DSA in our cohort, which is comparable with previous reports (preformed: 4.7–22.2% [13,19,32]; de novo: 8–19.9% [16,19,21,22]). Similar to other authors, the DSA MFI threshold of our laboratory is >1000 [14,18,22]. This cut-off value lies within the lower range of published thresholds (e.g., Musat et al. 300 [15], Koch et al. ≥ 1500 [20], Taner et al. ≥ 2000 [32], Kaneku et al. ≥ 5000 [21]). The distribution of antibody- classes in preformed DSA (Appendix A) showed a predominance of class I-DSA. However, the proportion of antibody classes in preformed DSA varies in previous reports, and at this point we cannot provide an adequate explanation for this finding. Out of 184 patients with preformed DSA, O’Leary et al. reported *n* = 84 (45.7%) class I-, and *n* = 50 (27.2%) class II-DSA [13]. As reported by Tamura et al. [33], 6/8 (75%) patients with preformed DSA presented solely class II-, and none presented with class I-DSA only. Vandevoorde et al. [19] reported 14 patients with preformed DSA, with *n* = 3 (21.4%) having class I- and *n* = 5 (35.7%) having class II-DSA alone.

As there is no clear evidence on the effects of DSA in LT, we planned our analysis under consideration of potential pitfalls. First, we eliminated confounding risk factors for poor transplant outcome, such as high-urgency, multivisceral, and retransplantations [34,35,36] to achieve more homogeneous recipient groups in terms of demographic and clinical parameters. In a second step, we performed a 1:1 propensity score matching and assigned for each DSA-positive LT recipient a DSA-negative control with a similar predicted survival by the means of the BAR score and its individual components. For our analysis, we used the BAR score, as it achieves the highest accuracy in predicting the outcome after LT compared to other score systems such as MELD, D-MELD (donor age multiplied by recipient MELD), and DRI [28,29,37]. It consists of six items and includes key LT survival predictors as donor, recipient, and graft factors and is therefore very convenient in daily use. While some scores such as the DRM (donor to recipient model, 13 items) and the SOFT (survival outcome following liver transplant; 18 items) seem to outperform the BAR score in terms of the long-term survival [38], the BAR score has the best composition of usability (number of items) and accuracy and was therefore used for our analysis [28].

We observed two known sex-related effects in the analysis. First, the DSA-positive group consisted of significantly more women (48.7%) than the matched DSA-negative control (29.2%). This observation was even more evident in LT recipients with preformed DSA (65.3% women). The result is in accordance with the published data [13,15,32] and explained by previous pregnancies as the main trigger for anti-HLA antibodies prior to solid organ transplantation [39,40,41,42].

The second observed sex-related effect refers to the lower rate of LTs due to alcoholic cirrhosis in the DSA-positive group. The main indication for LT in men is still alcoholic liver cirrhosis [43,44], and no publication so far had linked alcohol consumption in LT recipients with lower HLA antibody development. Hence, this result is more likely to reflect an epidemiological reality than an immunological origin.

Female donor-to-male recipient mismatch is another possible risk factor for graft loss [45]. However, in our analysis F-M donor-to-recipient mismatches seem not to be relevant factors in graft and patient survival. Female-to-male mismatch was more present in the DSA-negative cohort, which showed better organ and patient survival rates. Second, within the F-M-mismatched patients, only those with preformed DSA showed a significantly lower patient (*p* = 0.004) and graft survival (*p* = 0.001) compared to matched DSA-negative recipients after one year.

While we observed a higher rate of autoimmune liver diseases (AILD) and cryptogenic cirrhosis or NASH in the DSA-positive cohort, this was not found to have a negative impact on the outcome. In fact, in some studies AILD and cryptogenic cirrhosis/NASH have been shown to achieve a favorable graft and patient survival [46,47,48]. This also applied for the AILD patients in our study cohort (patient survival at one and five years: both 85.7%). DSA-positive AILD patients showed a non-significant inferior patient survival compared to the DSA-negative AILD patients at one and five years (both 76.9% vs. 100%, *p* = 0.156).

DSA-positive LT recipients showed significantly shorter CIT and WIT compared with the matched DSA-negative control. An increase in these parameters is associated with a longer postoperative stay [49] and inferior transplant outcomes [34,50,51]. In our observation cohort, however, the DSA-positive patients showed a significantly longer ICU- and postoperative stay and worse short-term survival compared to DSA-negative LT recipients. The longer hospital stay in our cohort was mostly related to a prolonged rejection treatment in DSA-positive patients. No relevant differences in liver function parameters were observed between the groups, though three parameters differed at distinct time points without clinical correlates. Possible explanations for this result may be that DSA have no direct effect on liver function parameters, or the effects of DSA presence cannot be measured by the means of such routine liver function tests. In both situations, the results support the theory of some authors that the commonly used laboratory markers of the liver function are of limited predictive value in the interpretation of the course and prognosis of liver grafts [52,53].

With regard to patient and organ survival, we observed a significant difference in the one-year patient survival between the DSA-positive group and the unmatched DSA-negative control (*p* = 0.003). After the group matching, this effect was less pronounced and marginally missed the significance threshold (*p* = 0.053). However, a subgroup analysis revealed a significant correlation between the detection of preformed DSA and inferior patient survival after LT (*p* = 0.008). Regarding the 3-year and 5-year patient and graft survival, no significant differences were observed, regardless of the matching. Similarly, O’Leary et al. showed an association between preformed DSA and patient death [13] and, more recently, Tamura et al. [33] observed an inferior 90-day survival in living-donor LT recipients with preformed DSA. In accordance to O’Leary et al. [13], patients with high MFI (≥ 5000) preformed DSA showed an even more pronounced inferior survival compared to matched DSA-negative individuals (*p* = 0.022).

Additionally, Kaneku et al. have described the compromising effect of de novo DSA on patient and graft survival in the first year after LT [21]. In contrast to these results, the one-year survival of LT recipients with de novo DSA showed comparable outcomes to the matched DSA-negative control in our study. In turn, some authors could not confirm a correlation between DSA and inferior outcomes after LT [19,20]. These differences in study results may be caused by heterogeneous patient cohorts (analyzing first LTs, re-LTs and living donor LTs together), varying sample sizes, the use of different DSA detection methods, as well as the lack of a uniform definition for ABMR in LT.

Herein, this study is among those with the largest number of cases analyzed and the most recent observation period. Furthermore, all the DSA detections were performed on the Luminex^®^ platform, which allows highly accurate DSA determination. Over the last decade, this method has shown its great value in antibody assessment compared to older techniques, such as the complement-dependent cytotoxicity test (CDC) or ELISA. Not only does it provide a higher sensitivity and specificity, but also a more detailed HLA antibody specification [9,54]. This has led to a broader understanding of DSA pathophysiology and immunohistology.

With respect to patient survival within the first year after LT, our findings strongly support the validity of the previously published results regarding the impact of DSA in LT. The rate of suspected rejections and performed biopsies, and especially histologically confirmed TCMR, was higher in the DSA-positive cohort than in the DSA-negative control group. 

Furthermore, a remarkable finding of this study is that DSA patients have been treated significantly more often for (suspected) rejections. Especially in cases with the unclear or steroid-resistant deterioration of liver function and equivocal or missing biopsies, rejection therapy was augmented on clinical suspicion in the presence of DSA. In addition, patients with preformed DSA were treated in a highly vulnerable post-LT phase (6 days vs. 9 days in the DSA-negative cohort; *p* = 0.003). Another major finding of this study is that sepsis was observed significantly more often in DSA-positive LT-recipients and was the main cause of death in this cohort (*p* = 0.046). Koch et al. recently described sepsis as a relevant cause of death in DSA-positive LT recipients [20]. All our 11 DSA-positive patients who succumbed to death due to sepsis received cytoreductive therapy consisting of either IVIGs, PP, ATG, Rituximab, or the combination of two or more. 

This result is of utmost importance in the controversial discussion on the role of DSA in LT. We observed that the initiation or augmentation of rejection therapies in DSA-positive LT recipients with unclear delayed/impaired liver function may result in increased mortality due to infectious complications. This effect seems most pronounced in the first year after LT, particularly due to the increased vulnerability of LT recipients in this phase after transplantation, whereas the mid-term and long-term results were comparable for DSA-negative LT recipients. 

Certainly, our study has limitations. First, the single-center, observational, retrospective study design has limitations that we are well aware of, so that our findings cannot be immediately extrapolated to all LTs as we excluded also high-urgency and re-transplantations. Nonetheless, this design ensured a relatively low heterogeneity regarding the clinical interpretation and therapeutic management of DSA detection. Second, the used MFI threshold of >1000 was at the lower end of previous publications. However, no international MFI threshold standard has been defined so far, and the MFIs in our analysis were considerably above the detection limit (4332 (1470–7336)).

Third, the used BAR score for matching is not yet an established and validated method. Despite the high predictive power of the BAR score regarding survival in LT, such matching can also cause a selection bias and may also explain the observation of reciprocal results in the long-term course of the examined matched collectives. Given the study’s retrospective design, the therapeutic decision-making for rejection treatment is hard to reconstruct: some rejection treatment decisions were performed on a “ex juvantibus” basis and rejection protocols also changed throughout the follow-up. Additionally, histological samples taken before 2017 were rarely stained for C4d and no reevaluation according to the Banff 2016 criteria was performed. Although a reevaluation of the histological samples may have provided deeper understanding of the pathology of DSA in LT, it would have no impact on the past decision making. An important point is that the fear of DSA, but not actual ABMR, led to more robust immunosuppression treatments, which caused increased rates of sepsis-related death. Due to methodological issues at the time period analyzed, it was not possible to provide further information about complement-binding DSA or DSA subclasses. We decided not to include data on HLA classes and subclasses in the analysis, as a further breakdown would only increase the probability of a type I error and dilute the focus of the manuscript.

On the other hand, important insights into the role of DSA in LT were gained. While the immunological component for inferior short-term outcome after LT delivered by DSA cannot be excluded, our observation implies that immunosuppressive (over-) treatment notably increased morbidity due to infectious complications. Apparently, the trigger to initiate or increase rejection therapies in DSA-positive patients with equivocal histological findings and/or unclear deterioration of liver function is dramatically low due to a lack of clear diagnostic and therapeutic guidelines. Thus, the interpretation of DSA in LT should be considered very carefully.

We therefore propose to warily investigate suspected ABMR histologically before the initiation or augmentation of rejection treatment. Future steps need to be the development of a common consensus regarding MFI thresholds and simplified ABMR guidelines to better identify patients who may benefit from DSA detection and ABMR treatment in the context of LT.

## Figures and Tables

**Figure 1 jcm-09-03986-f001:**
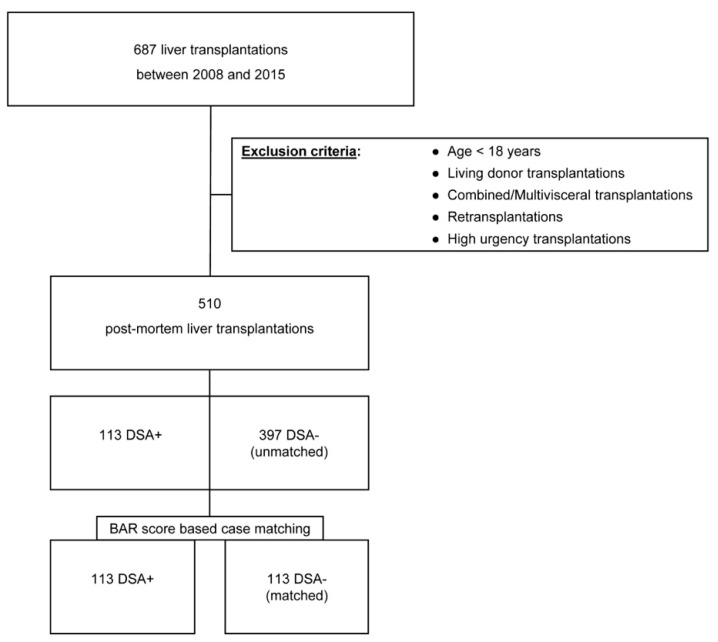
Visualization of the patient selection and matching process.

**Figure 2 jcm-09-03986-f002:**
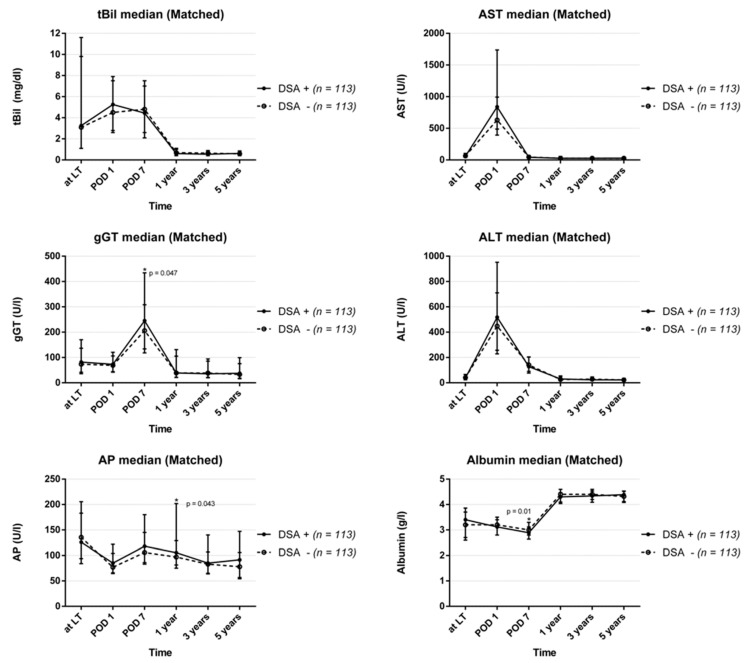
Course of routine laboratory parameters after liver transplantation divided into DSA-positive and DSA-negative recipients (total bilirubin: tBil; alanine aminotransferase: ALT; serum aspartate aminotransferase: AST; gamma-glutamyl transferase: gGT; albumin: ALB; alkaline phosphatase: AP).

**Figure 3 jcm-09-03986-f003:**
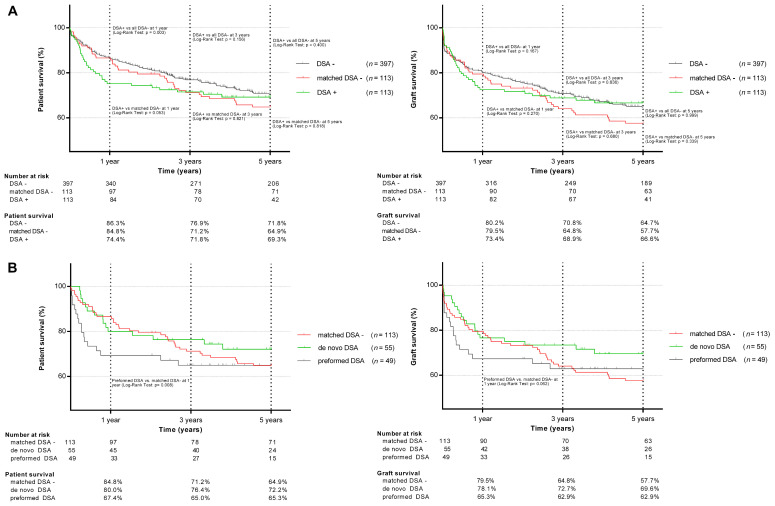
Patient and graft survival in DSA-positive, DSA-negative, matched DSA-negative (**A**), and preformed and de novo DSA (**B**) liver transplant recipients.

**Table 1 jcm-09-03986-t001:** Epidemiological, clinical, and operative data of patients who underwent LT from deceased donors between 1 January 2008, and 31 December 2015, sorted by DSA-positive versus all DSA-negative recipients (unmatched).

Variables	DSA+ (*n* = 113)		All DSA- (*n* = 397)		*p*-Value ^a^
**General**					
Recipient sex male, *n* (%)	58	(51.3)	286	(72.0)	**<0.001**
Recipient age (years)	54	(48–61)	57	(51–62)	0.073
Recipient BMI (kg/m^2^)	26.7	(24–31)	26.6	(24–30)	0.623
Donor sex male, *n* (%)	62	(54.9)	214	(53.9)	0.856
Donor age (years)	59	(45–70)	57	(44–70)	0.576
BAR score	8	(3–13)	6	(3–9)	**0.013**
DRI	2.5	(2.1–2.8)	2.5	(2–2.8)	0.288
labMELD	16	(9–22)	15	(10–20)	0.377
labMELD ≥ 35, *n* (%)	20	(17.7)	37	(9.3)	0.645
Gender mismatch F-M, *n* (%)	20	(17.7)	111	(28)	**0.028**
**Etiology, *n* (%)**					
Hepatitis C	20	(17.7)	77	(19.4)	0.685
Alcohol	34	(30.1)	188	(47.4)	**0.001**
PSC/PBC/AIH	13	(11.5)	33	(8.3)	0.296
NASH/cryptogenic cirrhosis	22	(19.5)	39	(9.8)	**0.005**
Metabolic/congenital	9	(8)	13	(3.3)	**0.030**
Other	17	(15)	54	(13.6)	0.696
Malignancy	37	(32.7)	161	(40.6)	0.133
Hepatocellular carcinoma	32	(28.3)	157	(39.5)	**0.029**
Cholangiocellular carcinoma	5	(4.4)	4	(1.0)	**0.015**
**Perioperative Characteristics**					
Time on waiting list (days)	155	(27–407)	109	(33–259)	0.158
Cold ischemia time (h)	9.7	(8–11)	9.6	(8–11)	0.760
Warm ischemia time (min)	43	(38–50)	45	(40–55)	0.082
Length of stay after LT (days)	38	(28–62)	29	(21–47)	**0.001**
Length of stay on ICU (days)	13	(7–32)	9	(6–15)	**<0.001**
Retransplantations, *n* (%)	6	(5.3)	34	(8.6)	0.268

Annotations: Data presented as *n* (%) or median and interquartile range (Q1–Q3). ^a^ Group comparisons: (1) categorical data: Pearson chi-square test. (2) Continuous variables: Mann-Whitney U test. Abbreviations: AIH: Autoimmune hepatitis; BAR: Balance of Risk Score; BMI: Body Mass Index; d: days; DRI: Donor Risk Index; DSA: Donor-Specific Antibodies; FFP: Fresh Frozen Plasma; Gender mismatch F-M: female donor-to-male recipient mismatch; h: hours; ICU: Intensive Care Unit; labMELD: laboratory Model of End-stage Liver Disease Score; LT: Liver Transplantation; min: minutes; NASH: Non-Alcoholic Steatohepatitis; PBC: Primary Biliary Cholangitis; PSC: Primary Sclerosing Cholangitis; RBC: Red Blood Cells.

**Table 2 jcm-09-03986-t002:** Epidemiological, clinical, and operative data of patients who underwent LT from deceased donors between 1 January 2008 and 31 December 2015 sorted by DSA-positive (DSA+) versus matched DSA-negative recipients (matched DSA-).

Variables	DSA+ (*n* = 113)		Matched DSA- (*n* = 113)		*p*-Value ^a^
**General**					
Recipient sex male, *n* (%)	58	(51.3)	80	(70.8)	**0.003**
Recipient age (years)	54	(48–61)	57	(51–62)	0.061
Recipient BMI (kg/m^2^)	26.7	(24–31)	26.3	(23–30)	0.271
Recipient hypertension, *n* (%)	40	(35.4)	34	(30.1)	0.395
Recipient diabetes type II, *n* (%)	37	(32.7)	35	(31)	0.775
Donor sex male, *n* (%)	62	(54.9)	60	(53.1)	0.790
Donor age (years)	59	(45–70)	56	(44–69)	0.301
BAR score	8	(3–13)	7	(4–13)	0.619
DRI	2.5	(2.1–2.8)	2.4	(2–2.8)	0.532
labMELD	16	(9–22)	16	(10–29)	0.426
labMELD >35, *n* (%)	20	(17.7)	16	(14.2)	0.686
Gender mismatch F-M, *n* (%)	20	(17.7)	30	(26.5)	0.109
MFImax	4332	(1470–7336)	-	-	-
**Etiology, *n* (%)**					
Hepatitis C	20	(17.7)	20	(17.7)	1.000
Alcohol	34	(30.1)	56	(49.6)	**0.003**
PSC/PBC/AIH	13	(11.5)	9	(7.9)	0.369
NASH/cryptogenic cirrhosis	22	(19.5)	11	(9.7)	0.058
Metabolic/congenital	9	(7.9)	5	(4.4)	0.270
Other	17	(15)	11	(9.7)	0.226
Malignancy	37	(32.7)	47	(41.6)	0.169
Hepatocellular carcinoma,	32	(28.3)	45	(39.8)	0.068
Cholangiocellular carcinoma	5	(4.4)	2	(2.7)	0.249
**Perioperative Characteristics**					
Time on waiting list (days)	155	(27–407)	101	(30–226)	0.052
Pretransplant mechanical, ventilated or organ-perfusion support, *n* (%)	6	(5.4)	4	(3.6)	0.322
Cold ischemia time (h)	9.7	(8–11)	10.2	(9–12)	**0.021**
Warm ischemia time (min)	43	(38–50)	48	(40–56)	**0.012**
RBCs given during LT (units)	8	(4–12)	6	(4–13)	0.301
RBCs given within 24 h after LT (units)	2	(0–4)	2	(0–4)	0.985
FFPs given during LT (units)	21	(16–32)	21	(16–30)	0.697
FFPs given within 24 h after LT (units)	5	(2–8)	4	(2–9)	0.369
Length of stay after LT (days)	38	(28–62)	32	(23–49)	**0.012**
Length of stay on ICU (days)	13	(7–32)	9	(6–15)	**0.008**
Retransplantations, *n* (%)	6	(5.3)	11	(9.7)	0.207
**One-year Mortality, *n* (%)**	29	(25.7)	18	(15.9)	
Hepatic	4	(3.5)	3	(2.7)	0.788
Cardiovascular	7	(6.2)	9	(8.0)	0.692
Infection/Sepsis	11	(9.7)	2	(1.8)	**0.046**
Malignancy	4	(3.5)	3	(2.7)	0.788
Other/Unknown	3	(2.7)	1	(0.9)	0.567

Annotations: Data presented as *n* (%) or median and interquartile range (Q1–Q3). ^a^ Group comparisons: (1) categorical data: Pearson chi-square test. (2) Continuous variables: Mann-Whitney U test. Abbreviations: AIH: Autoimmune hepatitis; BAR: Balance of Risk Score; BMI: Body Mass Ind ex; d: days; DRI: Donor-Risk-Index; DSA: Donor-Specific Antibodies; FFP: Fresh Frozen Plasma; Gender mismatch F-M: female donor-to-male recipient mismatch; ICU: Intensive Care Unit; h: hours; labMELD: laboratory Model of End-stage Liver Disease Score; LT: Liver Transplantation; min: minutes; MFImax: Median Fluorescence Intensity of the highest-ranked positive bead in the bead panel; NASH: Non-Alcoholic Steatohepatitis; PBC: Primary Biliary Cholangitis; PSC: Primary Sclerosing Cholangitis; RBC: Red Blood Cells.

**Table 3 jcm-09-03986-t003:** Reasons for one-year graft loss and mortality in preformed DSA, de novo DSA, and matched DSA-negative (matched DSA-) LT organ recipients ^a^.

	Preformed DSA (*n* = 49)		De Novo DSA (*n* = 55)		*p*-Value		Matched DSA- (*n* = 113)	*p*-Value ^b^	*p*-Value ^c^
Total graft loss and death	16	(14.2)	11	(9.7)		18	(15.9)		
Hepatic	2	(1.8)	1	(0.9)	1.782	3	(2.7)	0.732	0.566
Cardiovascular	5	(4.4)	2	(1.8)	0.446	9	(8.0)	0.268	0.087
Infection/Sepsis	7	(6.2)	3	(2.7)	0.384	2	(1.8)	**0.031**	0.264
Malignancy	1	(0.9)	3	(2.7)	0.131	3	(2.7)	0.347	0.494
Other/Unknown	1	(0.9)	2	(1.8)	0.332	1	(0.9)	0.933	0.279

Annotations: Data presented as *n* (%). ^a^ Two deaths observed in patients with both preformed and de novo DSA are not shown (1 hepatic, 1 infectious). ^b^ Comparison with organ recipients with preformed DSA only. ^c^ Comparison with organ recipients with de novo DSA only. Abbreviations: DSA: Donor-Specific Antibodies; IVIG: Intravenous Immunoglobulins; LT: Liver Transplantation

**Table 4 jcm-09-03986-t004:** Comparison of rejections and rejection treatments in LT organ recipients within the first postoperative year sorted by DSA-positive (DSA+) versus DSA-negative (DSA-) and matched DSA-negative (matched DSA-) recipients.

	DSA+ (*n* = 113)		All DSA- (*n* = 397)		*p*-Value ^a^	Matched DSA- (*n* = 113)		*p*-Value ^a^
**Cases with performed biopsies**	**72**	**(63.7)**	127	(32.0)	**<0.001**	35	(31.0)	**<0.001**
**Treated ABMR**	41	(36.3)	7	(1.8)	**<0.001**	2	(1.8)	**<0.001**
Biopsy proven	2	(1.8)	0	(0.0)	**0.008**	0	(0.0)	0.155
On clinical suspicion	39	(34.5)	7	(1.8)	**<0.001**	2	(1.8)	**<0.001**
**Treated TCMR**	71	(62.8)	135	(34.0)	**<0.001**	37	(32.7)	**<0.001**
Biopsy proven	59	(52.2)	118	(29.7)	**<0.001**	34	(30.1)	**<0.001**
On clinical suspicion	12	(10.6)	17	(4.3)	**0.010**	3	(2.7)	**<0.001**
**Rejection therapy**	77	(68.1)	135	(34.0)	**<0.001**	37	(32.7)	**<0.001**
Steroids ^b^	70	(61.9)	133	(33.5)	**<0.001**	37	(32.7)	**<0.001**
Plasmapheresis/IVIGs	41	(36.3)	7	(1.8)	**<0.001**	2	(1.8)	**<0.001**
Thymoglobulin	29	(25.7)	17	(4.3)	**<0.001**	3	(2.7)	**<0.001**
Rituximab	1	(0.8)	0	(0.0)	0.061	0	(0.0)	0.316
Other	16	(14.2)	41	(10.3)	0.254	13	(11.5)	0.551
Time from LT to therapy (days)	9.5	(6–17)	9	(7–23)	0.159	9	(7–23)	0.214

Annotations: Data presented as *n* (%) or median and interquartile range (Q1–Q3). Group comparisons: (1) categorical data: Pearson chi-square test. (2) Continuous variables: Mann-Whitney U test. ^a^ Comparison with DSA+ organ recipients. ^b^ Combined with an increase in standard immunosuppression. Abbreviations: ABMR: Antibody-mediated rejection; DSA: Donor-Specific Antibodies; IVIG: Intravenous Immunoglobulins; LT: Liver Transplantation; TCMR: T-cell-mediated rejection.

**Table 5 jcm-09-03986-t005:** Comparison of rejections and rejection treatments in LT organ recipients within the first postoperative year sorted by patients with de novo DSA versus preformed DSA and matched DSA-negative (matched DSA) recipients.

	Preformed DSA (*n* = 49)		De Novo DSA (*n* = 55)		*p*-Value	Matched DSA- (*n* = 113)		*p*-Value ^a^	*p*-Value ^b^
**Cases with performed biopsies**	**25**	**(51.0)**	39	(70.9)	**0.037**	35	(31.0)	**0.015**	**<0.001**
**Treated ABMR**	16	(32.7)	20	(36.4)	0.691	2	(1.8)	**<0.001**	**<0.001**
Biopsy proven	1	(2.0)	1	(1.8)	0.934	0	(0.0)	0.128	0.151
On clinical suspicion	15	(30.6)	19	(34.5)	0.670	2	(1.8)	**<0.001**	**<0.001**
**Treated TCMR**	26	(53.1)	38	(69.1)	0.093	37	(32.7)	**0.015**	**<0.001**
Biopsy proven	19	(38.8)	35	(63.6)	**0.011**	34	(30.1)	0.279	**<0.001**
On clinical suspicion	7	(14.3)	3	(5.5)	0.127	3	(2.7)	**0.005**	0.359
**Rejection therapy**	29	(59.2)	39	(70.9)	0.210	37	(32.7)	**0.002**	**<0.001**
Steroids ^b^	25	(51.0)	38	(69.1)	0.060	37	(32.7)	**0.028**	**<0.001**
Plasmapheresis/IVIGs	16	(32.7)	20	(36.4)	0.691	2	(1.8)	**<0.001**	**<0.001**
Thymoglobulin	9	(18.4)	18	(32.7)	0.095	3	(2.7)	**<0.001**	**<0.001**
Rituximab	1	(2.0)	0	(0.0)	0.287	0	(0.0)	0.128	
Other	5	(10.2)	9	(16.4)	0.358	13	(11.5)	0.809	0.381
Time from LT to therapy (days)	6	(5–16)	11.5	(7–18)	**0.024**	9	(7–23)	**0.003**	0.693

Annotations: Data presented as *n* (%) or median and interquartile range (Q1–Q3). Group comparisons: (1) categorical data: Pearson chi-square test. (2) Continuous variables: Mann-Whitney U test. Patients with both preformed and de novo DSA are not shown (*n* = 9). ^a^ Comparison with organ recipients with preformed DSA. ^b^ Comparison with organ recipients with de novo DSA. ^c^ Combined with an increase in standard immunosuppression. Abbreviations: ABMR: Antibody-mediated rejection; DSA: Donor-Specific anti-HLA Antibodies; IVIG: Intravenous Immunoglobulins; LT: Liver Transplantation; TCMR: T-cell-mediated rejection.

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
