# Peer review of "Treatment of Anti-HLA Donor-Specific Antibodies Results in Increased Infectious Complications and Impairs Survival after Liver Transplantation"

_jcm, 2020, doi:10.3390/jcm9123986_

Round 1

Reviewer 1 Report

The work sheds light on an underdeveloped topic in liver transplantation.
The background should be expanded to explain the importance of this work in daily transplant practice.
It is necessary to demonstrate that the donor-recipient woman-man matching, more present in the DSA + group, does not influence the outcome analyzed.
It is necessary to explain that the rate of liver disease related to autoimmune or cryptogenic diseases (31% in the DSA + group, 17.7% in the DSA- group) does not influence the outcome analyzed.
The discussion and conclusions are adequate. It is necessary to broaden the discussion on the Luminex-assay technique to make it credible.

Author Response

Reviewer 1:
1) The work sheds light on an underdeveloped topic in liver transplantation. The background should be expanded to explain the importance of this work in daily transplant practice.
Response: Thank you for the remark. In order to emphasize the importance of our work and the need for further evaluation of the clinical role of DSA, we added a more detailed, yet short reflection of previous study results on this topic. To explain the recently evolving course of further interest and investigation in this field, we also inserted a concise summary of the immunological background of DSA in ABMR (pages 1-2, lines 43-53). We are pleased to contribute to the clarification of the role of DSA in liver transplantation.
2) It is necessary to demonstrate that the donor-recipient woman-man matching, more present in the DSA+ group, does not influence the outcome analyzed.
Response: Thank you for this important comment. Gender mismatch is a known risk factor for poor graft survival after liver transplantation. Female-to-male (F-M) mismatches show the worst outcomes. In our analysis 28% (111/397) of DSA-negative patients showed a female-to-male donor-to-recipient (F-M) mismatch compared to 17.7% (20/113) in the DSA-positive group (p=0.028). In the matched cohorts we observed 26.5% F-M mismatches in the DSA-negative group (30/113, p=0.109 compared to DSA-positive). We included this information in Tables 1 (p. 5, “Gender mismatch”) and 2 (p. 6-7).
Within all patients in the matched cohorts, there was no significant difference in graft loss between patients with F-M-mismatch (n=50) and those without F-M-mismatch (n=176) in the first (78% vs. 75%, p=0.559) and third year after LT (64% vs. 65.3%, p=0.892) regardless of the DSA status. No significant difference could be observed between DSA-positive and matched DSA-negative F-M-mismatched patients in regard to patient survival after one, three and five years [70% vs. 86,7%, p=0.268; 65% vs. 70%, p=0.546; 60% vs. 63.3%, p=595]. The most pronounced effect on graft survival within F-M-mismatched patients was observed in patients with preformed DSA (n=6) compared to the DSA-negative recipients after one (33.3% vs. 83.3%, p=0.004) and three years (33.3% vs. 63.3%, p=0.047). This also applied for patient survival in these groups (1y: 33.3% vs. 86.7%, p=0.001 and 3y: 33.3% vs. 70%, p=0.014).
Albeit F-M mismatch is a possible risk factor for graft loss (Meta-analysis by Lai et al., PMID: 29853738) other known donor risk factors need to be considered. We therefore included the Donor Risk Index (DRI) in our analysis to rule out major confounding donor-related risk factors. Since the F-M-mismatch is not included in
2
the DRI, we now added a section in the results (p. 4, l. 158-160; p. 8, l. 193-194) and discussion (p. 14, l. 323-328) to clarify this matter for the reader.
Considering the results mentioned above, we conclude, that the F-M mismatch has no significant impact on the result in our study cohort for two reasons: First, the fact that the DSA-positive group was the one with the lowest rate of F-M-mismatches but inferior graft and patient survival rates. Second, within the F-M-mismatched patients, preformed DSA were the main risk factor for impaired patient survival.
3) It is necessary to explain that the rate of liver disease related to autoimmune or cryptogenic diseases (31% in the DSA + group, 17.7% in the DSA- group) does not influence the outcome analyzed.
Response: Thank you for this interesting consideration. Autoimmune liver diseases (AILD) as indications for LT have shown excellent survival rates in Europe (PMID: 22609307) and the USA (PMID: 22117641). Moreover, due an often prolonged treatment with corticosteroids and a certain rate of AILD recurrence, the impact on long-term outcome seems to be more pronounced than on short-term outcome (PMID: 28706981, PMID: 22117641).
Patients with cryptogenic cirrhosis (CC) have similar graft and patient survival rates compared to NASH, alcoholic cirrhosis and AILD (PMID: 29215462). Considering non-alcoholic fatty liver disease (NAFLD) and AILD as probable underlying etiology of CC (PMID: 20030578, PMID: 26983869), this could be one explanation for the similar survival rate.
This observation also applied for our matched cohort with a one/three-year organ survival of 81 / 81% in AILD and 75.1 (p=0.547) / 63.4% (p=0.143) in non-AILD LTs. Patient survival after one, three and five years was 85.7 / 85.7 / 85.7% in AILD and 78.5 (p=0.436) / 68.3 (p=0.122) / 63.9% (p=0.07) in non-AILD recipients, and therefore even better in patients transplanted for AILD than non-AILD. For AILD, patient survival in DSA-positive compared to DSA–negative patients was non-significantly inferior after one and five years (both 76.9% vs. 100%, p=0.156).
Patient survival after one, three and five years was 67.7 / 67.7 / 67.7% in patients with CC or NASH compared to 81 (p=0.083) / 70.3 (p=0.583) / 65.6% (p=0.885) for all the other entities. Organ survival was 71 / 67.7% vs. 76.4 (p=0.512) / 64.6% (p=0.905). Out of the patients transplanted for CC/NASH, DSA-positive patients showed a slightly higher patient survival rate at one and five years (both 72.7% vs. 63.6%, p=0.367).
In this study, we observed a more pronounced effect of DSA on short-term graft and patient survival and this has already been shown in other studies. There were no significant differences in graft/patient survival between DSA-positive and –negative patients transplanted for either AILD or NASH/CC. Consequently, there is no evidence accounting for a negative impact of the higher rate of AILD and NASH/CC in the DSA-positive group. We added this issue into our discussion (p. 14, l. 330-336).
4) The discussion and conclusions are adequate. It is necessary to broaden the discussion on the Luminex-assay technique to make it credible.
Response: Thank you for the remark. We added a short summary of the advantages of the luminex-assay to underline our methodology in the discussion section (p. 15, l. 372-376).

Reviewer 2 Report

The present study assess the impact of DSA on patient and graft survival by  means of a matched case-control analysis and  the aim is to identify risk factors for inferior patient and graft outcome.
The topic is of interest; the paper is well written. The tables are neat and clear.
However, there are few points that need to be addressed:

- It would be of interest to know the mean MFI in preformed and de novo DSA as some studies suggest that high MFI preformed DSA correlate with a poor prognosis and lower liver graft survival.  Does the author have information on C1q or C3-binding DSA as this can also have a different impact on graft survival?  Interestingly in the present study preformed DSA were preponderant of class I and not class II as in previous studies (O’Leary[15] and Tamura[44]), how can the author explain?

- What was the average time untill the first detection of de novo DSA ?

-Consider to detail the site and type of infections (nosocomial, opportunistic, multidrug-resistant organism etc), the time occurrence (early or late >6 months) prior or after rejection treatment  and what type of prophylactic antimicrobials were used in the author’s center with special focus on patients treated for rejection. Please consider to include this  data in analysis.

-Consider to discuss the performance of  BAR score as presented by De Boer et al. De Boer JD, Putter H, Blok JJ, Alwayn IPJ, Van Hoek B, Braat AE. Predictive Capacity of Risk Models in Liver Transplantation. Transplant Direct. 2019;5(6):1-11. doi:10.1097/TXD.0000000000000896 . Overall graft survival is best predicted by the DRM (donor to recipient model) or SOFT (survival outcome following liver transplant ). Consider that BAR score best predicts short term survival and not long term survival. Discuss why authors choose this propensity match score over the other scores.

Author Response

Reviewer 2:
1) The present study assesses the impact of DSA on patient and graft survival by means of a matched case-control analysis and the aim is to identify risk factors for inferior patient and graft outcome. The topic is of interest; the paper is well written. The tables are neat and clear. However, there are few points that need to be addressed: It would be of interest to know the mean MFI in preformed and de novo DSA as some studies suggest that high MFI preformed DSA correlate with a poor prognosis and lower liver graft survival. Does the author have information on C1q or C3-binding DSA as this can also have a different impact on graft survival?
Response: We thank the reviewer for this comment. For preformed DSA, the median MFImax was 4836 (IQR 2077-7645) vs. 2326 (1096-6272) for de novo DSA (p=0.015). Out of the patients with preformed DSA, 38.8% (n=19) and 23.6% (n=13) of those with de novo-DSA (p=0.095) had a MFImax of ≥5000. Patients with preformed DSA with a MFImax ≥5000 showed an even more pronounced, significant difference between one-year patient survival (63.2% vs. 84.8%, p=0.022) compared to the matched DSA-negative group (p.10, Figure 3). We added this finding to our results (p. 10, l. 233-237) and discussion (p. 15, l. 359-361) For de novo-DSA with a MFImax ≥5000 no significant difference could be observed compared to the DSA-negative matches (92.3% vs. 84.1%, p=0.419).
Our results would therefore support the hypothesis of a possible correlation of high MFI preformed DSA with poor prognosis.
The Luminex-assay was launched in our laboratory in 2008. Complement-binding assays only evolved over the following years and were not applied for LT in a regular, standardized manner. Therefore, unfortunately we cannot provide information on C1q-, C3- or C4d-binding DSA for this retrospectively analyzed cohort. A note was inserted in the discussion to make this evident to the reader (p. 16, l. 416-417).
Furthermore, as the aim of the study was to investigate the general impact of the presence of DSA, we did not reevaluate previously collected sera using complement-binding assays.
2) Interestingly in the present study preformed DSA were preponderant of class I and not class II as in previous studies (O’Leary [15] and Tamura [44]), how can the author explain?
Response: Thank you for the remark. In previous studies, the proportion of antibody- classes in preformed DSA varies. For example, out of 184 patients with preformed DSA, O’Leary et al. reported n=84 (45.7%) class I-, and n=50 (27.2%) class II-DSA (PMID: 23780820). As reported by Tamura et al. (PMID: 30991451), 6/8 (75%) patients with preformed DSA presented solely class II-, and none presented with class I-DSA only. Vandevoorde et al. (PMID: 29665189) reported of 14 patients with preformed DSA, with n=3 (21.4%) having class I- and n=5 (35.7%) having class II-DSA alone.
In our study cohort 32 (65.3%) patients with preformed DSA (n=49) had class I, and only 5 (10.2%) had class II-DSA. These results are similar to those by O’Leary et al., albeit the total sample size was more than three times greater. Since the other two publications mentioned only present a very small number of cases, it is questionable if these results can be taken into account to determine a trend in DSA-classes for LT. Considering the expression of HLA-class I in a wide population of human organs and cell types, it can be assumed that preformed anti-HLA antibodies consist of more class I-antibodies.
The increased expression of HLA-class I antigens in acute rejection of liver allografts (PMID: 3541328) could be a possible explanation for a more rapid evolution of antibody-mediated organ injury. However, the upregulation of HLA class II through previous activation of class I-HLA (PMID: 2204660) in states of liver injury might also play an important role in posttransplant ABMR due to preformed antibodies.
Therefore, more studies and greater sample sizes are needed to evaluate this important aspect. We addressed this in the discussion (p. 13, l. 288-295).
3) What was the average time until the first detection of de novo DSA?
Response: We thank the reviewer for this comment. The median time to the first detection of de novo-DSA in our cohort was 13 days (IQR 9.5-20). We included this result into the results section (p. 4, l. 148-149).
4) Consider to detail the site and type of infections (nosocomial, opportunistic, multidrug-resistant organism etc), the time occurrence (early or late >6 months) prior or after rejection treatment and what type of prophylactic antimicrobials were used in the author’s center with special focus on patients treated for rejection. Please consider to include this data in analysis.
Response: We thank the reviewer for this remark. Due to the retrospective character of this analysis it is very difficult to perform a precise workup of the infections. However, all death-related infections were early infections and occurred within the first 6 months after LT. Albeit we observed a clear correlation between the rejection treatments and the occurrence of infections, it is difficult to break down the infection to a clear point in time. Some patients already received antimicrobial treatment prior to rejection therapy, while others developed the infection days to weeks after the rejection treatment. Unfortunately, it is not possible to clearly differentiate this from our data. The most common infection-related cause of death was sepsis due to exacerbation of hospital-acquired pneumonia (n=7 in DSA+; and n=2 in DSA- patients). In three DSA+ cases the origin of the sepsis was not clear (30%). We included this data in the manuscript (p. 9, l. 215-217). Antibiotic prophylaxis before rejection therapy is not a standard in our clinic and was therefore not applied. In case of infection, antibiotic therapy was administered in a targeted manner according to the antibiogram.
5) Consider to discuss the performance of BAR score as presented by De Boer et al. De Boer JD, Putter H, Blok JJ, Alwayn IPJ, Van Hoek B, Braat AE. Predictive Capacity of Risk Models in Liver Transplantation. Transplant Direct. 2019;5(6):1-11. doi:10.1097/TXD.0000000000000896. Overall graft survival is best predicted by the DRM (donor to recipient model) or SOFT (survival outcome following liver transplant). Consider that BAR score best predicts short term survival and not long term survival. Discuss why authors choose this propensity match score over the other scores.
Response: We thank the reviewer for this important remark. The reviewer refers to an important publication which investigates the predictive power of different survival scores in liver transplantation (PMID:
31321293). De Boer et al. recently confirmed that the BAR score has a very good predictive power for short term survival. Albeit its predictive power for long-term survival is also significantly better compared to other tests, the DRM and SOFT scores were superior. We absolutely agree that this relevant finding needs to be addressed in our manuscript.
For the decision to use the BAR score in our analysis we investigated many publications. In 2011 Dutkowski el. al introduced the BAR score and showed its superiority compared to other available scores and handiness (only 6 items) by including key LT survival predictors as donor, recipient and graft factors. Schlegel et al. additionally showed that of all models, only the BAR score was linearly associated with complications and was most useful for risk classification in liver transplantation, based on expected posttransplantation mortality and morbidity (PMID: 22042468; PMID: 21618688; PMID: 27676319). In all these publications, both short-term and long-term survival were significantly better displayed by the BAR score than by other scores.
However, the mentioned publication consists of very recent data and therefore unfortunately did not receive any attention in the planning and realization of our study. Furthermore, although we are aware of the good predictive power of the SOFT score the disadvantages outweighed the decision to use this score for matching in our analysis: The score consists of a total of 18 items and is therefore complex and less suitable for everyday use compared to the BAR score. We addressed and discussed this decision in the discussion section of the manuscript (p. 14, l. 303-310)

Round 2

Reviewer 2 Report

All comments have been addressed.